# Disrupted Mitochondrial Network Drives Deficits of Learning and Memory in a Mouse Model of *FOXP1* Haploinsufficiency

**DOI:** 10.3390/genes13010127

**Published:** 2022-01-11

**Authors:** Jing Wang, Gudrun A. Rappold, Henning Fröhlich

**Affiliations:** 1Department of Human Molecular Genetics, Institute of Human Genetics, Heidelberg University Hospital, Im Neuenheimer Feld 366, 69120 Heidelberg, Germany; jing.wang@med.uni-heidelberg.de (J.W.); gudrun.rappold@med.uni-heidelberg.de (G.A.R.); 2Interdisciplinary Center for Neurosciences, Im Neuenheimer Feld 366, 69120 Heidelberg, Germany

**Keywords:** FOXP1 syndrome, *Foxp1*^+/−^ mouse, hippocampus, associative learning, autism spectrum disorder, mitochondrial dysfunction, reactive oxygen species

## Abstract

Reduced cognitive flexibility, characterized by restricted interests and repetitive behavior, is associated with atypical memory performance in autism spectrum disorder (ASD), suggesting hippocampal dysfunction. FOXP1 syndrome is a neurodevelopmental disorder characterized by ASD, language deficits, global developmental delay, and mild to moderate intellectual disability. Strongly reduced Foxp1 expression has been detected in the hippocampus of *Foxp1*^+/−^ mice, a brain region required for learning and memory. To investigate learning and memory performance in these animals, fear conditioning tests were carried out, which showed impaired associative learning compared with wild type (WT) animals. To shed light on the underlying mechanism, we analyzed various components of the mitochondrial network in the hippocampus. Several proteins regulating mitochondrial biogenesis (e.g., Foxo1, Pgc-1α, Tfam) and dynamics (Mfn1, Opa1, Drp1 and Fis1) were significantly dysregulated, which may explain the increased mitophagy observed in the *Foxp1*^+/−^ hippocampus. The reduced activity of complex I and decreased expression of Sod2 most likely increase the production of reactive oxygen species and the expression of the pre-apoptotic proteins Bcl-2 and Bax in this tissue. In conclusion, we provide evidence that a disrupted mitochondrial network and the resulting oxidative stress in the hippocampus contribute to the altered learning and cognitive impairment in *Foxp1*^+/−^ mice, suggesting that similar alterations also play a major role in patients with FOXP1 syndrome.

## 1. Introduction

Autism spectrum disorder (ASD) and intellectual disability (ID) frequently co-occur and have overlapping risk genes [1,2]. The forkhead box protein P1 (*FOXP1*) gene (OMIM 605515) is one of several hundred genes that have been associated with ASD and ID [3,4,5]. *FOXP1* haploinsufficiency causes FOXP1 syndrome, a neurodevelopmental disorder manifesting in autistic traits, ID, speech and language deficits and dysmorphic features [6,7]. Homozygous loss-of-function variants in *FOXP1* have not been described in humans and very likely lead to embryonic death, as in *Foxp1*^−/−^ mice [8].

FOXP1 belongs to an evolutionarily ancient and highly conserved protein subfamily comprising four members, FOXP1 to FOXP4. All four transcription factors act as dimers, both homo- and hetero-dimers. For example, Foxp1 forms hetero-dimers with Foxp2 and Foxp4 [9,10]. In the brain, FOXP1 expression is restricted to specific areas such as the cortex (layer III–VIa), striatum, thalamic nuclei, and the CA1/CA2 region of the hippocampus [11,12].

Mitochondrial dysfunction contributes to the pathology of various neurodevelopmental disorders and so may play a role in FOXP1 syndrome. Some characteristics of FOXP1 syndrome are suggestive of hippocampal damage, but the exact mechanisms are still unknown. In this study, we investigated whether mitochondrial alterations occur in the hippocampus of *Foxp1*^+/−^ mice and whether these alterations underlie mitochondrial dysfunction.

## 2. Materials and Methods

### 2.1. Animals

Mice were kept in the specific pathogen-free Interfacultary Biomedical Facility (IBF) at the University of Heidelberg under a 12-h light–dark cycle, and given *ad libitum* access to water and food. All procedures were conducted in compliance with the National Institutes of Health Guidelines for the Care and Use of Laboratory Animals and were in accordance with the German Animal Protection Law (TierSCHG). The day of birth was considered as postnatal day (P) 0.5. The behavioral experiment was approved by the Governmental Council Karlsruhe, Germany.

### 2.2. Fear Conditioning

The experiment was performed as described before [13]. For acquisition, the mice spent 6 min inside a conditioning chamber (Med Associates Inc. St. Albans, VT, USA) and were exposed to an acoustic signal (conditioned stimulus). At the last second of each tone segment (5000 Hz, 85 dB, 30 s), one footshock (0.5 mA, 1 s) was applied via the floor grid (unconditioned stimulus). After 24 h, the animals were re-introduced to the chamber for 6 min to evaluate contextual fear memory without tone and footshock. Cue-related fear memory was assessed for 4 min in the remodeled chamber with two 30 s conditioned stimuli after 48 h of training. Freezing behavior was analyzed via a video camera connected to video tracking software (Video Freeze®, Med Associates Inc., St. Albans, VT, USA).

### 2.3. RNA Isolation and cDNA Synthesis

Total RNA was prepared from frozen mouse hippocampal tissue samples using peqGOLD TriFast™ (PEQLAB-Life Science, Radnor, PA, USA). First-strand cDNA synthesis was performed from 1.5 μg total RNA using a Superscript II reverse transcriptase kit (ThermoFisher Scientific, Waltham, MA, USA) and oligo dT12-18-primers (ThermoFisher Scientific, Waltham, MA, USA) according to the manufacturer’s instructions.

### 2.4. Quantitative Real-Time PCR

Quantitative real-time PCR was performed using the qTOWER system (Analytic Jena, Jena, Germany) with an annealing temperature of 60 °C using SYBR Green No-ROX Fast Mix (Bioline, Luckenwalde, Germany) according to the manufacturer’s instructions. Each sample was analyzed in triplicate and relative mRNA levels were assessed using the Standard Curve Method, normalized to succinate dehydrogenase complex subunit A (*Sdha1*) and hypoxanthine phosphoribosyltransferase 1 (*Hprt1*). The primer sequences are listed in Appendix A.

### 2.5. Calculation of Mitochondrial DNA Copy Number

The analysis was performed as previously described [14]. In brief, the expression of three mitochondrial genes, *16sRNA, D-loop*, and *Nd1*, as well as the two nuclear genes, *B2m* and *Hk2*, was measured by quantitative real-time PCR and the copy number was determined by the mtDNA/nDNA ratio. The primer sequences are listed in Appendix A.

### 2.6. Protein Isolation and Analysis

Total protein lysates were isolated from frozen mouse hippocampal tissue samples using RIPA buffer with cocktail protein inhibitors (ThermoFisher Scientific, Waltham, MA, USA). Proteins (30–60 μg) were separated on a 12% SDS-PAGE gel and detected by primary antibodies (Appendix A) according to the manufacturer’s instructions. An anti-Gapdh mouse monoclonal/rabbit polyclonal antibody was used as a control. IRDye 800CW and IRDye 680 (LICOR Biosciense, Lincoln, NE, USA) were used as secondary antibodies (Appendix A). Protein bands were visualized using the Odyssey Infrared Imaging System and quantified using Image Studio Lite 4.0 software (both LI-COR Biosciences, Lincoln, NE, USA).

### 2.7. Isolation of Mitochondria

Freshly dissected tissue was homogenized in ice-cold isolation buffer (20 mM HEPES, 250 mM sucrose, 10 mM KCl, 1.5 mM MgCl_2_, 1 mM EGTA, 1 mM EDTA), supplemented with cocktail protease inhibitors (ThermoFisher Scientific, Waltham, MA, USA). After differential centrifugation, the mitochondria-enriched pellet was cleaned with washing buffer (250 mM sucrose, 5 mM HEPES, 5 mM KOH, 0.1 mM EGTA, pH 7.2).

### 2.8. Analysis of Complex I Activity

The activity of complex I in hippocampal tissue was determined using the Complex I Enzyme Activity Microplate Assay Kit (#ab109721, Abcam, Cambridge, UK) according to the manufacturer’s instructions.

### 2.9. Quantification of Reactive Oxygen Species (ROS)

Hippocampal ROS were marked by 2′,7′-dichlorodihydrofluorescein diacetate (Molecular Probes™ H2DCFDA (H2-DCF, DCF), ThermoFisher Scientific, Waltham, MA, USA) as previously described [15] and assessed by a DS-11 Series spectrophotometer/fluorometer (DeNovix Inc., Wilmington, DE, USA).

### 2.10. Analysis of Cytochrome c Release

Mitochondrial and cytosolic protein isolation was performed as previously described [16]. Cytochrome c release was determined by calculating the ratio of occurrence in the mitochondria to cytoplasm. Cytochrome c, Cox IV (a marker for mitochondrial proteins), and Gapdh (a marker for cytosolic proteins) were evaluated by western blot.

### 2.11. Statistics

Investigators were blinded to genotypes, and animals were randomly assigned to tests. Primary data were analyzed using SPSS Statistics 25.0.0 (IBM, Armonk, NY, USA) and Microsoft Office Excel (Microsoft Corporation, Redmond, WA, USA). Outliers were identified by SPSS Statistics 25.0.0 (IBM, Armonk, NY, USA), strong outliers (≥3 standard deviations above the mean) were excluded from further analysis. All data were tested for normal distribution using the Kolmogorov–Smirnov and Shapiro–Wilk tests. Two-way ANOVA was performed with litter as the second factor. *p* values ≤ 0.05 were considered significant.

## 3. Results

### 3.1. Foxp1^+/−^ Mice Exhibit Impaired Associative Learning and Memory

Contextual and cued fear conditioning measures the ability of a mouse to learn and remember an association between an aversive experience and environmental cues. *Foxp1*^+/−^ mice showed a significantly lower percentage of freezing than WT animals did in the 6 min training phase, which included four cue-shock pairings. Overall, average freezing during training was reduced by ~67% in these animals (Figure 1). In *Foxp1**^+/−^* animals, average freezing in contextual fear conditioning without tone and foot shock was reduced by ~70% during the 6 min period and it was reduced by ~73% in the testing of cue-related fear memory over 4 min after 48 h (Figure 1). These results indicate that *Foxp1**^+/−^* animals have deficits in associative learning.

### 3.2. Foxp1 Deficiency in the Hippocampus Leads to Altered Mitochondrial Biogenesis

Within the forebrain, Foxp1 is strongly expressed in the pyramidal neurons of the CA1/CA2 hippocampal subfields. Ventral CA1 hippocampal projections to the basal amygdala encode conditioned fear memory after exposure to aversive stimuli [17]. We quantified *Foxp1* mRNA expression in the hippocampus of WT and *Foxp1*^+/−^ animals at three developmental stages (P1.5, P12.5, and adult) by quantitative real-time PCR as well as protein expression in adult animals by western blot. *Foxp1* mRNA levels were significantly reduced in P1.5, P12.5, and adult *Foxp1*^+/−^ tissue by ~44%, ~41%, and ~28%, respectively (Figure 2A). This also applied to both hippocampal Foxp1 isoforms, Foxp1 A and D, which were decreased by ~51% and ~29%, respectively (Figure 2A). Foxp1 is a transcription factor and may affect genes involved in mitochondrial biogenesis [18,19], so we analyzed different members of this pathway. Decreased *Foxo1*, *Pgc1α*, and *Tfam* mRNA expression was detected in the adult *Foxp1*^+/−^ hippocampus, with protein levels reduced by ~29%, ~23%, and ~29%, respectively (Figure 2B–D).

As mitochondrial biogenesis affects the mitochondrial DNA (mtDNA) copy number, we also examined the ratio of mtDNA to nuclear DNA (nDNA) at all three stages in WT and *Foxp1*^+/−^ animals. *Foxp1*^+/−^ hippocampi displayed significantly reduced mtDNA/nDNA ratios at P1.5 and decreased ratios in P12.5 and adult tissue (Figure 2E).

### 3.3. Disrupted Mitochondrial Dynamics and Increased Mitophagy in the Foxp1^+/−^ Hippocampus

Mitochondrial dynamics, consisting of coordinated cycles of fusion and fission, is an essential process to maintain mitochondrial shape, distribution, and size upon changing conditions. Two proteins instrumental in outer and inner mitochondrial membrane fusion are mitofusin-1 (Mfn1) and dynamin-like 120 kDa protein (Opa1). In contrast, the recruitment of cytoplasmic dynamin-1-like protein (Drp1) and its anchoring in mitochondria by mitochondrial fission protein 1 (Fis1) is required for fission. Mfn1 protein was significantly reduced by ~39% in the adult *Foxp1*^+/−^ hippocampus compared with WT. This is also the case for both Opa1 isoforms, L- and S-Opa1 (~42% and ~35%, respectively), resulting in a decreased L-Opa1/S-Opa1 ratio (Figure 3A). In contrast, both Drp1 and Fis1 proteins showed significantly elevated levels, by ~20% and ~60%, respectively (Figure 3B). Our observations suggest the impaired maintenance of outer and inner mitochondrial membrane fusion and elevated recruitment of Drp1 to mitochondria with increased fission in the *Foxp1*^+/−^ hippocampus.

Fis1 also promotes mitophagy, a process by which damaged or excess mitochondria are removed. To study autophagy and autophagic cell death, we determined the amount of microtubule-associated protein 1A/1B light chain 3I (LC3I) and LC3II by western blot. LC3II is a structural autophagosomal membrane protein that is formed by conjugation of LC3I to phosphatidylethanolamine. Our analysis revealed significantly increased LC3A/BII expression in the adult *Foxp1*^+/−^ hippocampus compared with WT tissue (Figure 3C). A pathway involving Pink1 kinase and Parkin ubiquitin ligase is crucial for the removal of damaged mitochondria. Indeed, the expression of Pink1 and Parkin protein is significantly elevated in the *Foxp1*^+/−^ hippocampus, by ~31% and ~67%, respectively (Figure 3D,E).

### 3.4. Foxp1^+/−^ Hippocampi Show Increased Oxidative Stress, Cytochrome c Release, and Expression of Apoptosis Markers

The maintenance of neuronal homeostasis and function depends on proper mitochondrial biogenesis, dynamics, and mitophagy. Disrupted mitochondrial function can increase reactive oxygen species (ROS) and lead to cell death. As our results indicate mitochondrial damage, we quantified ROS in hippocampal tissue using 2′,7′-dichlorofluorescein (DCF). ROS were significantly increased in *Foxp1*^+/−^ tissue compared with WT tissue (Figure 4A). However, mitochondrial dysfunction and redox imbalance may be related or occur independently. In fact, we observed a ~14% reduced activity of mitochondrial complex I in adult *Foxp1*^+/−^ tissue (Figure 4B). Moreover, an important constituent in oxidative signaling, the antioxidant superoxide dismutase 2 (Sod2), is a direct transcriptional target of Foxo1 [20] and was significantly reduced in the *Foxp1*^+/−^ hippocampus (Figure 4C).

Increased oxidative stress can lead to apoptosis [21,22]. Therefore, we studied the expression of Bcl2 and Bax, which are important regulators of apoptotic cell death. Consistent with our previous findings, increased Bcl-2 and Bax levels in the *Foxp1*^+/−^ hippocampus indicated an increased number of pre-apoptotic cells (Figure 4D). The mitochondrial intermembrane space contains several proteins promoting cell death, such as cytochrome c. After apoptosis is initiated, Bax changes its conformation, migrates to the mitochondrial membrane, and mediates the release of cytochrome c from the intermembrane space into the cytosol. Cytochrome c expression was not different between the cytoplasmic or mitochondrial fraction in adult WT and *Foxp1*^+/−^ tissue. However, *Foxp1*^+/−^ hippocampi showed a significant increase in the ratio of cytoplasmic to mitochondrial cytochrome c compared with WT hippocampi (Figure 4E).

These results suggest that mitochondrial damage associated with oxidative stress in the hippocampus contributes to the cognitive impairments in *Foxp1*^+/−^ animals.

## 4. Discussion

Studying brain regions involved in specific deficits of a neurodevelopmental disorder and understanding their molecular causes of damage are the basis for developing specific and efficient treatments. We have previously demonstrated that total lack of Foxp1 in the nervous system leads to severe behavioral abnormalities such as impaired social behavior and short-term memory, anxiety, and hyperactivity. In addition, morphological disruption of the striatum and the hippocampus have been detected and electrophysiological properties of hippocampal pyramidal neurons investigated [23]. Similarly, *Foxp1*^cKO^ mice lacking Foxp1 in the pyramidal neurons of the neocortex and the CA1/CA2 subfields of the hippocampus display a significant reduction in both the entire hippocampal volume and individual hippocampal subfields [24]. In addition, *Foxp1*^cKO^ animals exhibit behavioral defects associated with autism such as decreased sociability, hyperactivity, and anxiety.

ASD is characterized by reduced sociability, communication, and the occurrence of repetitive behaviors and restricted interests. One of the underlying causes is decreased attention and associative learning. This also holds true for attention deficit hyperactivity disorder (ADHD) and ID, which are both common comorbidities in FOXP1 syndrome [7,25]. In the present study, we investigated associative learning in murine *Foxp1* haploinsufficiency using contextual and cued fear conditioning tests, which is one of the most widely used paradigms to assess learning and memory. *Foxp1*^+/−^ mice showed significant deficits in both context-evoked and cue-evoked fear memory recall. Contextual and cued fear conditioning depends on the proper functioning of the amygdala, hippocampus, frontal cortex, and cingulate cortex. The amygdala plays an important role in both contextual and auditory stimuli [26]. Input from the hippocampus, specifically the dorsal hippocampus and CA3 region, is needed for contextual learning, whereas this input is not specifically needed for cue association learning [27,28]. Foxp1 is an important factor in cortical radial migration as well as neuronal morphogenesis in the developing cerebral cortex, and the somatic activity of cortical pyramidal neurons was shown to be directly involved in fear acquisition and extinction [29,30]. Therefore, we consider it likely that cortical damage also plays a role in the observed dysfunctions.

Remarkably, it has been shown that striatal circuits also play a role in fear conditioning. Both animals lacking Foxp1 in striatal D1 and D2 medium spiny neurons showed significant deficits in contextual and cognitive fear conditioning, with reduced freezing in both tests [31]. Mitochondrial dysfunction and increased oxidative stress have also been described in the *Foxp1*^+/−^ striatum [32] underlying mitochondrial dysfunction, and we hypothesize that similar dysfunctions may also occur in the *Foxp1*^+/−^ hippocampus.

Genes essential in the regulation of mitochondrial biogenesis and metabolism (Foxo1, Pgc-1α, Tfam) were downregulated in the *Foxp1*^+/−^ hippocampus. In particular, Pgc-1α, a direct transcriptional target of Foxo1 [33], represents a transcriptional coactivator and master regulator of mitochondrial biogenesis [19,34]. Its downregulation most likely leads to the observed reduction in Tfam in *Foxp1*^+/−^ tissue, a protein that regulates mitochondrial genome replication and transcription [35,36], and explains the decreased mitochondrial DNA copy number.

Pgc-1α is also essential to balance fusion and fission [37]. Both Mfn1 and Opa1 are required for the fusion of the outer and inner mitochondrial membranes, while the recruitment of cytoplasmic Drp1 and its anchoring in mitochondria by Fis1 is required for fission. The observation of decreased Mfn1 and Opa1 levels and increased Drp1 and Fis1 expression in the adult *Foxp1*^+/−^ hippocampus strongly suggests the impaired maintenance of outer and inner mitochondrial membrane fusion and increased fission. Disturbed mitochondrial dynamics therefore may lead to an altered structure and increased fragmentation, which trigger cellular stress.

In addition to proper mitochondrial biogenesis and dynamics, damaged mitochondria must be effectively removed to ensure mitochondrial function. Interestingly, elevated Fis1 was described as playing a role in mitochondrial fragmentation and autophagosome formation [38]. Indeed, increased LC3A/BII levels in *Foxp1*^+/−^ tissue indicate enhanced autophagy as these microtubule-associated proteins are necessary for autophagosome assembly [39]. This is also supported by the increased expression of Pink1 and Parkin, two proteins that promote the degradation of ROS-producing mitochondria, known as mitophagy [40].

Similar to the findings in the *Foxp1*^+/−^ striatum, the increased ROS levels in the *Foxp1*^+/−^ hippocampus are caused by a decreased complex I activity with increased ROS production, and a decreased degradation due to reduced Sod2 expression, a direct transcriptional target of Foxo1 [20]. On the one hand, increased oxidative stress in cells can damage mitochondrial respiration. Primary striatal *Foxp1*^+/−^ neurons after 8 days of differentiation did not show alterations in their oxygen consumption rates [32]. However, accumulating ROS with age may eventually damage mitochondrial respiration and affect ATP synthesis in *Foxp1*^+/−^ animals. On the other hand, increased oxidative stress in cells may ultimately lead to apoptosis. Indeed, the observed changes in cytochrome c release as well as increased Bax and Bcl2 expression may suggest this. Whether increased ROS accumulation results in increased cell death or rather leads to cell damage and cellular functional deficits or whether there are morphological changes of the neurons remains to be seen.

In summary, we demonstrate that mitochondrial dysfunction and oxidative stress in the hippocampus are most likely key contributors to the impaired associative learning in *Foxp1*^+/−^ animals. Thus, we expect that improving associative learning by targeting dysregulated proteins involved in mitochondrial function, electron transport chains, or antioxidants may have a positive impact on both the symptoms of ASD and ID in patients with FOXP1 syndrome.

## Figures and Tables

**Figure 1 genes-13-00127-f001:**
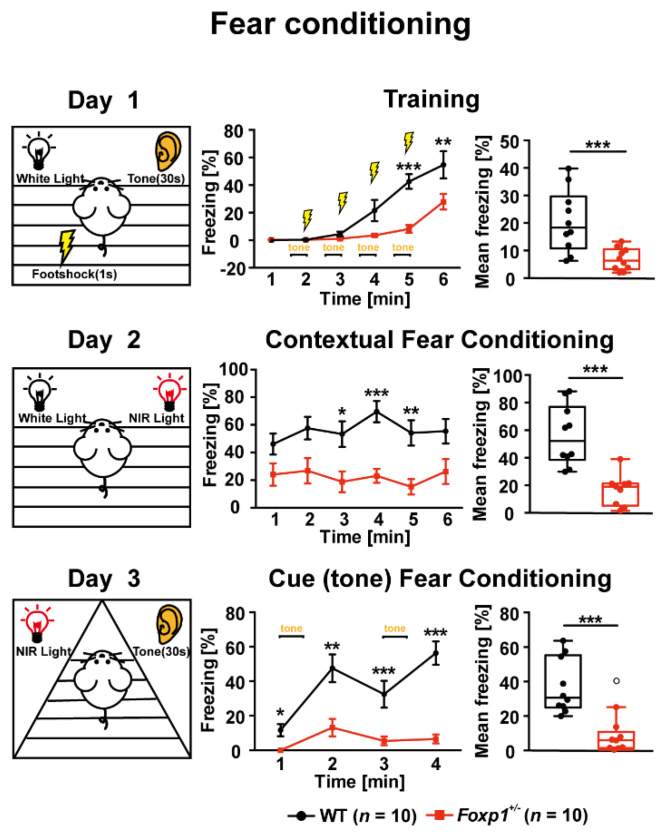
Fear-associated learning and memory is disrupted in *Foxp1*^+/−^ animals. In the 6 min acquisition phase (training, day 1), after the first tone/foot shock, *Foxp1*^+/−^ mice showed significantly less freezing compared to their WT littermates. Overall, mean freezing was reduced by ~67%. This was also true for contextual fear conditioning without tone and foot shock (day 2), and cued fear conditioning (day 3). In these experiments, the average freezing of *Foxp1*^+/−^ animals was decreased by ~70% and ~72%, respectively. At least ten animals were used per group. Diagrams display mean values ± SEM. In the box-and-whisker plot depicting average freezing, the boxes represent the first (lower box) and third quartiles (upper box), the whiskers represent 95% confidence intervals, and the lines within the boxes are median values. Weak outliers are marked with a circle. Asterisks indicate a significant difference (* *p* ≤ 0.05, ** *p* ≤ 0.01, *** *p* ≤ 0.001); two-way ANOVA; *n* = 10/10.

**Figure 2 genes-13-00127-f002:**
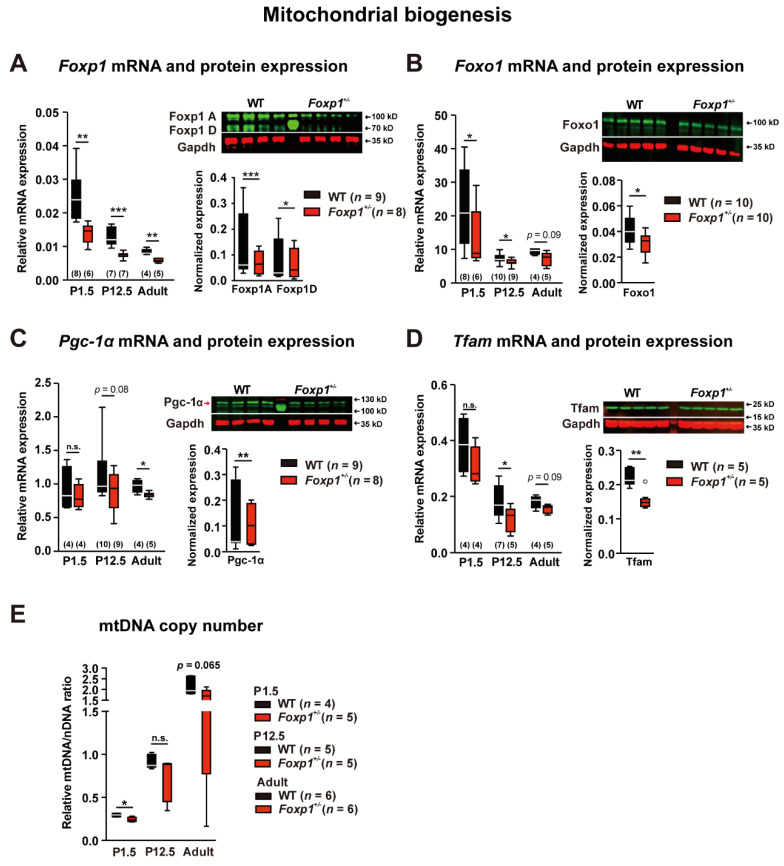
Impaired mitochondrial biogenesis in the *Foxp1*^+/−^ hippocampus. The mRNA levels of *Foxp1*, *Foxo1*, *Pgc-1α*, and *Tfam* were analyzed in WT and *Foxp1*^+/−^ tissue by quantitative real-time PCR at P1.5, P12.5, and adult stage. Protein expression of the four genes was quantified at adult stage by western blot (**A**–**D**). (**A**) *Foxp1* mRNA levels were significantly reduced, by ~40%, at all three time points. The two Foxp1 isoforms present in the hippocampus, Foxp1 A and Foxp1 D, were decreased by ~50% and ~29%, respectively, in *Foxp1^+/−^* tissue compared to WT. (**B**) *Foxo1* mRNA levels were decreased in *Foxp1^+/−^* mice at all three time points by ~40%, ~15% and ~25%, respectively and Foxo1 protein was reduced by ~29%. (**C**) At adult stage, *Pgc-1α* mRNA levels were ~14% lower and protein expression was reduced by ~23% in *Foxp1*^+/−^ tissue. (**D**) *Tfam* mRNA levels were reduced at P1.5, P12.5, and adult stages by ~20%, ~34%, and ~14%, and protein expression was reduced by ~29%. (**E**) To evaluate mitochondrial DNA copy number in WT and *Foxp1*^+/−^ tissues, mitochondrial DNA (mtDNA) and nuclear DNA (nDNA) were quantified by quantitative real-time PCR at P1.5, P12.5, and adult stages, and mtDNA/nDNA ratios were calculated. Reduced mtDNA/nDNA ratios in the *Foxp1*^+/−^ hippocampus at all three timepoints indicate a lower number of mitochondrial genomes per cell in *Foxp1*^+/−^ tissue compared to WT. The exact number of animals tested per group is indicated in the figure. Weak outliers are marked with a circle. Asterisks indicate a significant difference (* *p* ≤ 0.05, ** *p* ≤ 0.01, *** *p* ≤ 0.001); n.s. indicates non-significant; two-way ANOVA.

**Figure 3 genes-13-00127-f003:**
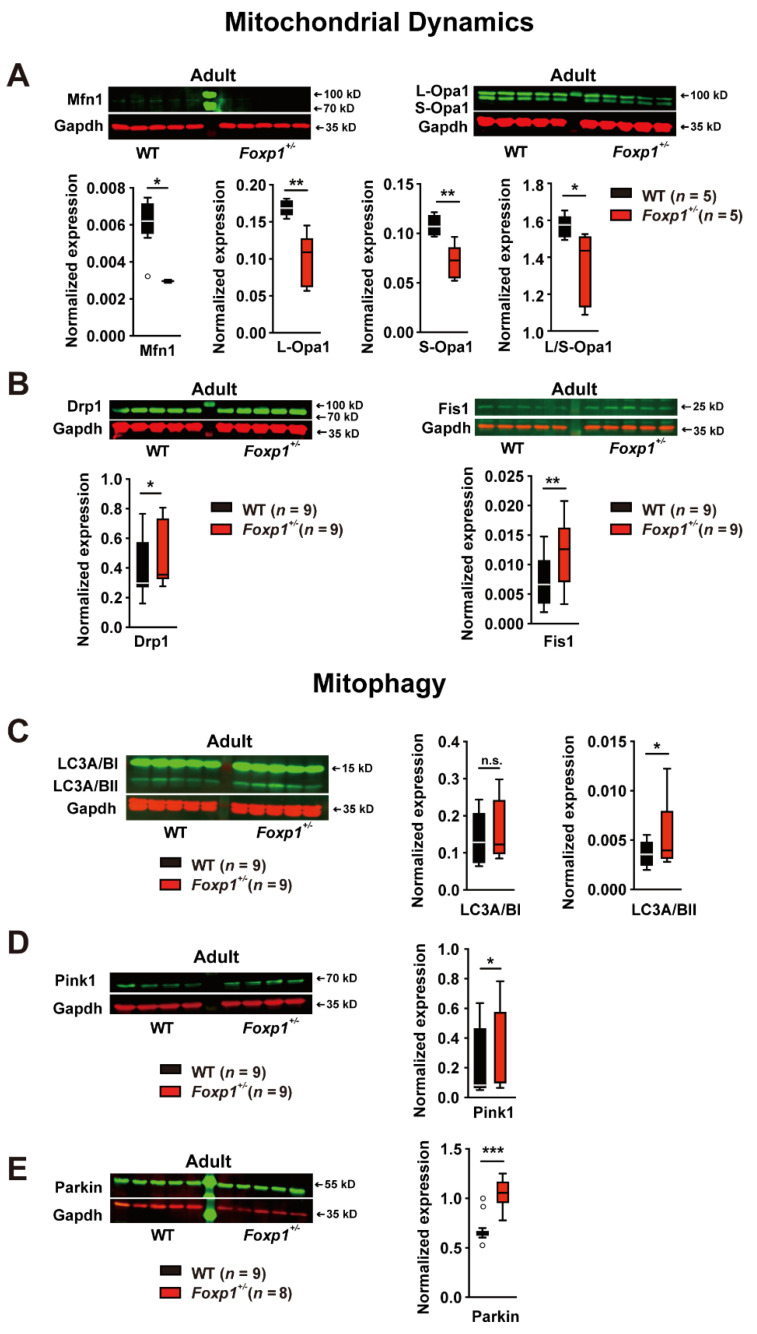
Altered mitochondrial dynamics and increased mitophagy in the adult *Foxp1*^+/−^ hippocampus. Mfn1 and Long (L)/short (S)-Opa1 regulating the fusion of the outer and inner mitochondrial membranes, and Drp1 and Fis1 promoting fission were quantified in the adult WT and *Foxp1*^+/−^ hippocampus by western blot (**A**,**B**). (**A**) Mfn1 was reduced by ~39% in *Foxp1*^+/−^ tissue, and L-Opa1 and S-Opa1 levels were decreased by ~42% and ~35%, respectively, leading to a lowered L-Opa1/S-Opa1 ratio by ~14% suggesting impaired fusion. (**B**) Increased expression of Drp1 and Fis1 by ~20% and ~60%, respectively, points to elevated fission in the *Foxp1^+/−^* hippocampus. (**C**) To evaluate autophagy, LC3A/BI and LC3A/BII abundance was quantified by western blot. Increased LC3A/BI and LC3A/BII levels in *Foxp1*^+/−^ tissue indicate an increased incidence of autophagy. (**D**,**E**) To investigate mitophagy, Pink1 and Parkin levels were determined by western blot. Both Pink1 and Parkin expression were elevated in the *Foxp1*^+/−^ hippocampus by ~31% (**D**) and by ~67% (**E**), respectively. The exact number of animals tested per group is indicated in the figure. Weak outliers are marked with a circle. Asterisks indicate a significant difference (* *p* ≤ 0.05, ** *p* ≤ 0.01, *** *p* ≤ 0.001); n.s. indicates non-significant; two-way ANOVA.

**Figure 4 genes-13-00127-f004:**
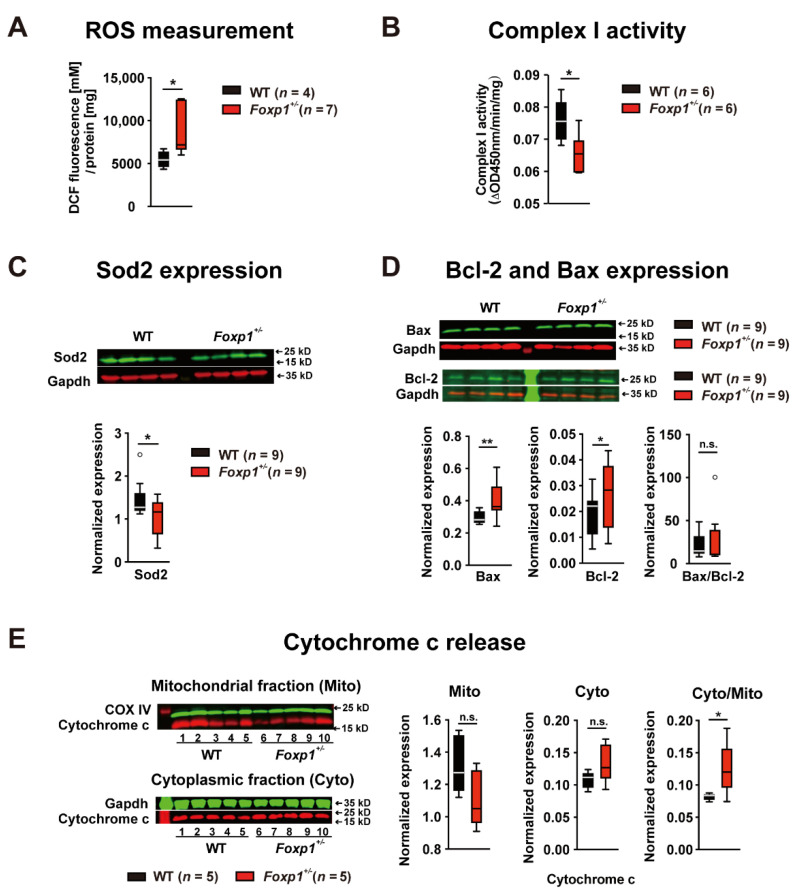
Adult *Foxp1*^+/−^ mice show increased oxidative stress and pre-apoptosis in the hippocampus. (**A**) Abundance of ROS as quantified by DCF staining. Increased ROS by ~67% in the *Foxp1*^+/−^ hippocampus indicates elevated oxidative stress. (**B**) Complex I enzyme activity as determined with the Microplate Assay Kit (Abcam). Complex I activity was decreased by ~14% in *Foxp1*^+/−^ tissue compared to WT. (**C**) Antioxidant enzyme Sod2 was quantified by western blot and showed a ~29% reduced expression in the *Foxp1*^+/−^ hippocampus. (**D**) Bax and Bcl-2, two markers of pre-apoptosis, were increased by ~71% and ~38% in *Foxp1*^+/−^ tissue, whereas the Bax/Bcl-2 ratio did not differ between genotypes. (**E**) The release of cytochrome c induced by certain pro-apoptotic stimuli was tested in WT and *Foxp1*^+/−^ tissues by quantifying the protein in mitochondrial and cytosolic fractions by western blot. A ~52% increased ratio of cytosolic to mitochondrial cytochrome c compared with WT indicates increased cytochrome c release in the *Foxp1*^+/−^ hippocampus. The exact number of animals tested per group is indicated in the figure. Weak outliers are marked with a circle. Asterisks indicate a significant difference (* *p* ≤ 0.05, ** *p* ≤ 0.01); n.s. indicates non-significant; two-way ANOVA.

## Data Availability

All data generated in this study are included in the manuscript.

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
