# Peer review of "Disrupted Mitochondrial Network Drives Deficits of Learning and Memory in a Mouse Model of *FOXP1* Haploinsufficiency"

_genes, 2022, doi:10.3390/genes13010127_

Round 1
Reviewer 1 Report
The manuscript “Disrupted mitochondrial network drives deficits of learning and memory in a mouse model of FOXP1 haploinsufficiency” examines the role of mitochondrial dysfunction in the pathology of Foxp1 syndrome. The results increase our understanding of the pathological mechanisms of ASD and ID. However, there is still some potential for improvement.
- The authors should provide the references for supporting the statement regarding the expression of Foxp1 in brain areas in lines 41-43.
- In line 259-260 the authors make a very bold statement and based on their results they can only speculate that mitochondrial damage associated with oxidative stress might be one of the mechanisms that contribute to the cognitive impairments in Foxp1+/- animals. The statement should be modified.
- The authors have not discussed about the mitochondrial respiration in the Foxp1+/- animals. The discussion should include their speculation about the overall rate of mitochondrial respiration in these animals in light of their findings.
- In the materials and methods section striatal tissue has been mentioned in place of hippocampus several times which should be corrected.
- In fig 3 the representative blot of Mfn1 does not represent a decrease in this protein. Also Mfn1 blot seems very similar to Drp1. This needs to be addressed.
Author Response
1. The authors should provide the references for supporting the statement regarding the expression of Foxp1 in brain areas in lines 41-43.
We have now included two relevant references: Ferland et al, 2003 and Co et al., 2020 (new references 11 and 12).
2. In line 259-260 the authors make a very bold statement and based on their results they can only speculate that mitochondrial damage associated with oxidative stress might be one of the mechanisms that contribute to the cognitive impairments in Foxp1+/- animals. The statement should be modified.
We have mitigated our statement according to the suggestion of the reviewer and deleted “strongly”.
3. The authors have not discussed about the mitochondrial respiration in the Foxp1+/- animals. The discussion should include their speculation about the overall rate of mitochondrial respiration in these animals in light of their findings.
Thank you for the suggestion, this is an interesting point. We now address this in the discussion (page 11, yellow highlighted section).
4. In the materials and methods section striatal tissue has been mentioned in place of hippocampus several times which should be corrected.
Thank you very much; it is now corrected.
5.In fig 3 the representative blot of Mfn1 does not represent a decrease in this protein. Also, Mfn1 blot seems very similar to Drp1. This needs to be addressed.
Thank you very much for pointing this out. Unfortunately, we have shown a wrong blot in Figure 3. Here, we now show the correct blot in which the expression differences are clearly visible. Please excuse this mistake.
Reviewer 2 Report
This is a well performed study showing the remarkable mitochondrial dysfunction in FOXP1 knockdown mice. Two minor concerns are: 1) it would be great to show the images of mitochondrial morphology of these mice in hippocampus and 2) the study of release of cytochrome c was not convincing while such biochemical assays were subject to possible artifacts. It will be great to providing imaging data to demonstrate the release of cytochrome c.
Author Response
1. it would be great to show the images of mitochondrial morphology of these mice in hippocampus
We agree with the reviewer that the study of mitochondrial morphology is an interesting and obvious point. However, we do not expect severe structural alterations, despite the behavioral abnormalities described. Foxp1+/- animals also do not show reduced life expectancy. Subtle changes in mitochondrial morphology on the other side may be masked by variance or fixation artifacts in EM analysis; therefore, such analysis would also need to be performed on very large numbers of animals and be very time consuming.
2. the study of release of cytochrome c was not convincing while such biochemical assays were subject to possible artifacts. It will be great to providing imaging data to demonstrate the release of cytochrome c.
Based on our results from the Foxp1+/- striatum (unpublished), we did not expect strong changes in cytochrome c release in the Foxp1+/- hippocampus. We therefore chose quantification by western blot, as we consider this method more accurate than imaging techniques.